# Stormwater Reservoir Sizing in Respect of Uncertainty

**Bartosz Szeląg [1],\*, Adam Kiczko [2]**  **and Lidia Dąbek [3]**

[1]  Department of Geotechnics and Water Engineering, Faculty of Environmental, Geomatic and Energy Engineering, Kielce University of Technology, 25-314 Kielce, Poland

[2]  Department of Hydraulics, Faculty of Civil and Environmental Engineering, Warsaw University of Life Sciences (WULS-SGGW), 02-787 Warsaw, Poland; adam_kiczko@sggw.pl

[3]  Department of Water and Wastewater Technology, Faculty of Environmental, Geomatic and Energy Engineering, Kielce University of Technology, 25-314 Kielce, Poland; ldabek@tu.kielce.pl

\*  Correspondence: bartoszszelag@op.pl

**Abstract:** The sizing of the stormwater reservoir, as the design of its properties, usually requires simulations of a basin runoff for a long rainfall series using a hydrodynamic model. In the case of insufficient observations, the rainfall series can be reproduced using empirical approaches. One of the crucial elements in the sizing of the stormwater reservoir is determination of duration time and intensity of rainfall (design rainfall event), for which the maximum reservoir capacity is being obtained. The outcome is, however, affected by significant uncertainty of runoff modeling. The aim of the study is to analyze the effect of the uncertainty of a rainfall-runoff model on calculated capacities of stormwater reservoirs, along with estimated duration times of the design rainfall. The characteristics of the rainfall events—intensity, duration, and frequency—were reproduced using an empirical approach of IDF (Intensity–Duration–Frequency). The basin response to the precipitation was modeled using the SWMM (Storm Water Management Model) and its uncertainty was estimated on the basis of the GLUE (Generalized Likelihood Uncertainty Estimation) method. The obtained probabilistic solution was compared with the deterministic one, neglecting the uncertainty. Duration times of the design rainfall determined in respect of the reservoir outflow using the probabilistic model were longer than those found with a deterministic approach. This has an effect on the desired capacities of the stormwater reservoir, which were overestimated when uncertainty was neglected.

**Keywords:** stormwater reservoir; Generalized Likelihood Uncertainty Estimation (GLUE); design rainfall event; Storm Water Management Model (SWMM)

## 1. Introduction

Intensive precipitation might result in inflows exceeding the capacity of stormwater drainage, causing flooding and releases of stormwater to the recipient (i.e., river). One of the possible solutions is attenuating stormwater discharges with storage reservoirs, built within sewer systems. The problem of designing these structures have attracted many researchers [1,2]. Their main focus are methods for finding the required reservoir capacity, ensuring proper reduction of flood flows. The problem can be also stated in respect of desired reduction of pollutant load in stormwaters [3].

The storage reservoirs are usually designed to operate as overflow reservoirs. Their main advantage is maintaining a constant outflow [4–6]. Many theoretical studies were devoted to methodology of designing this type of the reservoir [7–10]. Its capacity is usually determined for so-called design rainfall. Its duration is found to maximize required reservoir capacity, that allows for maintenance of the desired outflow [11]. The determination of the design rainfall duration

is usually done on the basis of many simplifications to the runoff model and description of the stochasticity of the rain events. In practice, it is done using nomograms of the rainfall duration and required reservoir capacity, elaborated for the given physio-geographic properties of the basin, drainage standards, and outflow devices [4,6,11]. Capacities obtained in such a graphical approach are considered to be affected with high uncertainty, because of very rough parametrization of the basin. A more detailed representation of the basin response to the rainfall can be obtained using hydrodynamic modeling [7,12–14]. In such an approach, reservoir capacities are calculated on the basis of so-called continuous simulations [2,7,12], where the input consists of long precipitation time series (around 30 years long). In the case where an observation period is significantly shorter than 30 years, available time series can be used to develop a synthetic precipitation generator that allows for extension of the series length artificially [15,16]. Recently, multidimensional probability density distributions parametrized with copula functions [17,18] are gaining popularity for precipitation generators, allowing high resolution of simulated time series to be obtained (below 5 min). In such an approach, in a precipitation time series, rainfall events are identified and then parametrized in a sense of height and duration, assuming usually the rectangular shape of the precipitation graph. On the basis of empirical distributions of these precipitation parameters, two dimensional functionals are being developed, which can be used in precipitation simulations. In other approaches, the dependency between the precipitation and duration is explained using the correlation coefficient, and then the rainfall events can be generated using the Iman-Conover method [19]. In many successful studies, duration and the rainfall height were considered as independent variables. One of the most sophisticated methods for a rainfall model includes canonical microsaccades, based on the fractal theory [12].

In practical studies, usually there is a lack of observed time series. The design rainfall can be determined using regional precipitation models, which on the basis of empirical relationships allows for estimation of its height for the given duration and occurrence frequency [20,21]. The approach takes a form of so-called Intensity–Duration–Frequency (IDF) curves and can be also used for a rainfall generator. Because the observation series of precipitation are rare in engineering applications, IDF approach is often used for the design of hydraulic structures in sewer systems [8,22].

Runoff models used in the design of the storage reservoirs are affected with strong uncertainty [23–26]. Surprisingly, its effect on the design of sewer system devices is, however, rarely analyzed. Kiczko et al. [27] showed that it significantly affects the determined capacities of the storage reservoir. This study included the design of the stormwater reservoir for a single rainfall event, using SWMM model. The model uncertainty was estimated using the well-known hydrology method of Generalized Likelihood Uncertainty Estimation (GLUE) [28]. However, the study focused on a single event, and did not address the problem of dependency of the design uncertainty on precipitation height and duration (in a result, the rainfall probability).

The aim of the present study was to develop a general method for the design of a stormwater reservoir in respect to the runoff model uncertainty, using the IDF approach for the design rainfall. It analyzes the effect of the model uncertainty for different probabilities of the design rain and shows its significance on the determination of duration times for the rainfall event that maximizes the reservoir capacity. As in the previous study, model uncertainty was assessed using the GLUE approach. Reservoir capacities obtained in respect of the model uncertainty were compared with those calculated neglecting it. The study utilizes IDF curves developed for Poland.

## 2. Materials and Methods

### 2.1. Object of Study

The analyzed catchment is located in the central-eastern part of Kielce and its area is 62 ha (Figure 1). The total area of the City Kielce is 109 km$^2$, with the population reaching 200,000 (density 21.4 people/ha). The highest point of the catchment is 271.20 m above sea level and the lowest one

is 260 m above sea level. The total length of the sewer network is 5583 m, of which the main canal is 1569 m long and its diameter from top to outlet varies between 600–1250 mm. The main channel receives rainwater from 17 side channels, whose diameters vary in the range of 300–1000 mm. The total volume of pipes with stormwater wells is 2032 m$^3$. The drop of the collector in individual sections varies from 0.04% to 3.90%, while the drop of side channels is 2.61% maximum. The area is covered with housing estates, public utility buildings, as well as main and side streets. The catchment contains 6 types of runoff areas: roofs (14.3%), pavements (8.4%), roads (17.7%), parking lots (11.2%), greenery (47.2%), and school pitches (1.3%). Road density in the basin is about 108 m/ha. More detailed data concerning the characteristics of the catchment are described by Dąbkowski et al. [29].

According to the DWA A-117 (2006) method, the annual number of rainfall events in the observation period (2008–2016) was around 36–58. The total rainfall height and its duration were varying in ranges of 3.0–45.2 mm and 20–2366 min, respectively. The duration of rainless periods was from 0.16 to 60 days. Annual precipitation heights in the observation period were changing between 537–757 mm, with rainfall days 155–266. The annual air temperature was 8.1–9.6 °C and the number of days with snowfall 36–84.

A stormwater treatment plant (STP) is located on the outflow from the catchment. At a distance of approximately 4.0 m from the distribution chamber (DC), an ultrasonic flow meter MES1 is installed in channel S1. The MES1 ultrasonic flow meter measures the filling and flow of stormwater in a 1-min step. Obtained time series of the discharge (MES1 point in Figure 1) were used for the identification of the runoff model.

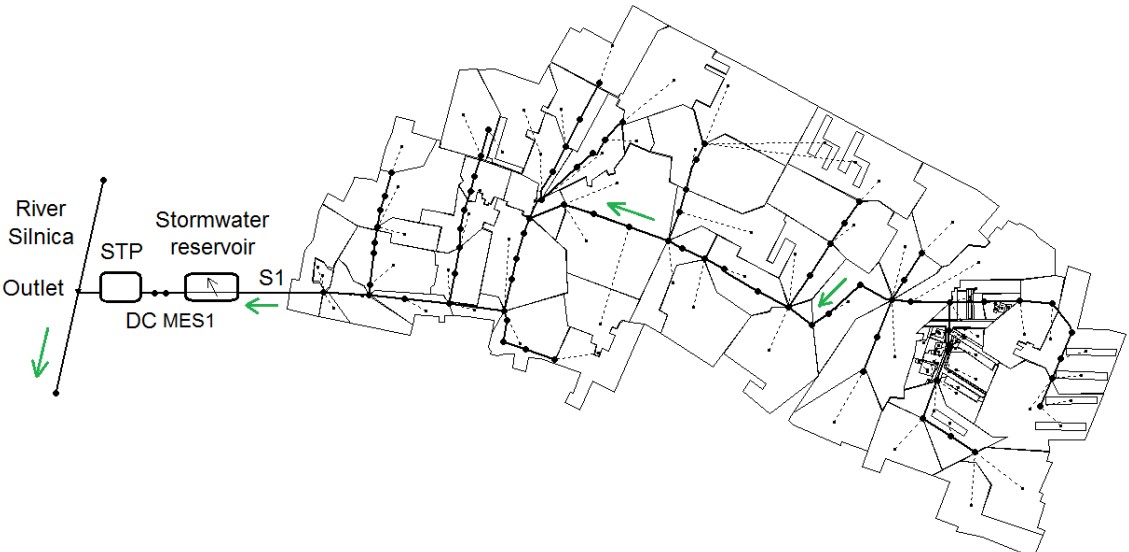

**Figure 1.** Location of the analyzed catchment and model diagram in Storm Water Management Model (SWMM) software.

The catchment, considered in operation, has already been the subject of numerous studies, in which surface runoff, stormwater quality, storm overflow, and stormwater treatment plant operation were modelled [3,16]. As part of these analyses, the model was calibrated using the trial and error method, where a set of parameters determining satisfactory results of calculations was searched for.

## 2.2. Methodology

In the absence of continuous, long-term rainfall data, the storage reservoir capacity is determined on the basis of the calculated rainfall frequency (*C* = 1, 2, 5, 10), using the category of drainage standard [11,30]. The calculation diagram of the developed method is presented in Figure 2.

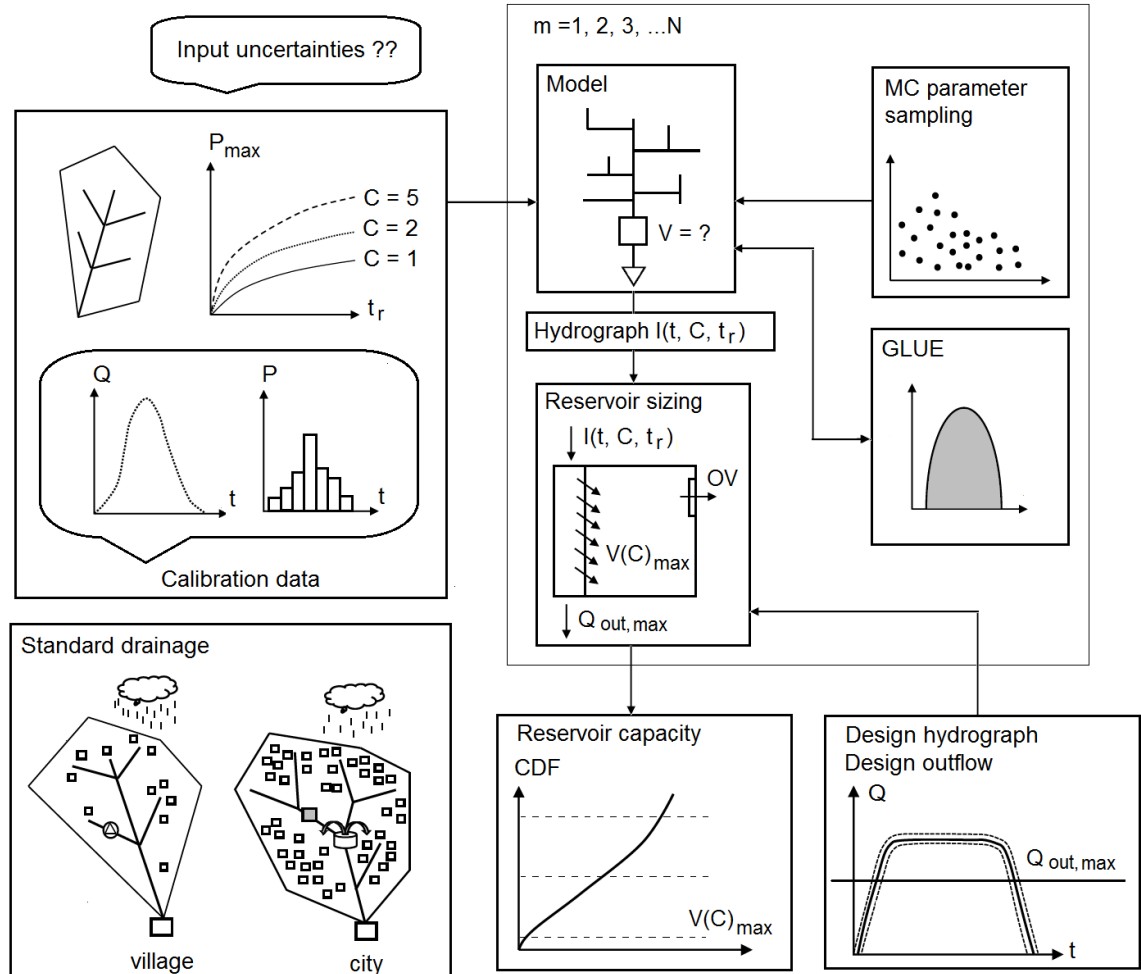

**Figure 2.** Calculation diagram of the method of dimensioning the retention capacity (where $I(t, C, t_r)$—reservoir inflow hydrograph for given rain duration time ($t_r$) and occurrence probability $C$, m = 1, 2, 3, ... , N—subsequent Monte Carlo simulations with varying parameters [$\alpha, d_{imp}, d_{perv}, n_{imp}, n_{perv}, n_n, \gamma, \beta$], for which reservoir inflow hydrograph is compute, $Q_{out,max}$—maximal allowed outflow form the reservoir, $V(C)_{max}$—maximal reservoir retention capacity for given values of $C$ and $Q_{out,max}$).

The computational data adopted in the study include the physical and geographical characteristics of the catchment and rainfall and flow measurements used to calibrate the hydrodynamic model. In order to determine the storage capacity of the reservoir, it is necessary to transform the calculated rainfall depth values with Equation (3) in the outflow values. In order to take into account the uncertainty of the catchment model, the Monte Carlo method is used to simulate the model parameters for the assumed ranges of their variability. Then, on the basis of measurements, the identification of distributions of analyzed parameters is performed with the GLUE method [31]. In this way the basin outflow hydrograph is determined, and using the balance of Equation (7), the reservoir capacity is calculated. In order to determine the maximum capacity of the reservoir for the assumed standard of drainage ($C$) and assumed outflow ($Q_{out,max}$), it is necessary to perform calculations for different ($m$) rainfall duration ($t_r$) in order to determine the extremum of variability of $V = f(C, t_r, Q_{out,max})$ and to determine the duration of the determining rain, where $V \rightarrow max\{(C, t_{rd}, Q_{out,max})\}$ (where: $t_{rd}$—design rainfall event). The values of the outflow from the retention reservoir ($Q_{out,max}$) and the determined accumulation capacities $V(C)_{max}$ were normalized by reference of the above variables to the impermeable area of the catchment ($F_{imp}$) and determination of the unit capacity index ($V_q$) and unit outflow ($q$).

### 2.3. Dimensioning the Retention Reservoir

A review of the literature [5–7,27] shows that the most common task of the retention reservoir in the stormwater system is to reduce the maximum flow at the outflow from the catchment and to improve the quality of the stormwater. At the reservoir design stage it is necessary to meet the condition that for the assumed return period at a given time period of rainfall (assumed repeatability of rainfall over a given period of time) and at its maximum filling $H = H_{max}$, the outflow value $Q_{out} \leq Q_{out,max}$ (design flow) does not lead to overloading of the stormwater treatment plant. Exceeding the $H_{max}$ value leads to an increase in outflow from the reservoir ($Q_{out} \geq Q_{out,max}$) and discharges with emergency overflow (OV, Figure 2). Therefore, the determination of the reservoir capacity is limited to the determination of the combination of $P_{max}$ and $t_r$ values, for which the desired capacity reaches the maximum value.

### 2.4. Uncertainty Analysis by the GLUE Method

Uncertainty analysis was performed using a method popular in hydrology: Generalized Likelihood Uncertainty Estimation (GLUE) [28,31]. It is based on the probabilistic formulation of the parameter identification problem, where instead of a single parameter set, as in the deterministic approach, a distribution of parameters is estimated. The uncertainty is modeled by the parameter variability. Because the output distribution is conditioned with the observations, a total model uncertainty is obtained. The method uses the Bayes formula:

$$P(Q/\Theta) = \frac{L(Q/\Theta) \cdot P(\Theta)}{\int L(Q/\Theta) \cdot P(\Theta) d(\Theta)} \tag{1}$$

where $P(\Theta)$ denotes *a priori* parameter distribution, $L(Q/\Theta)$ likelihood function, and $P(Q/\Theta)$ the *a posteriori* distribution, as the result of the parameter identification. In practical cases, the assumption on the *a priori* distribution is weak and usually takes a form of the uniform distribution limited to the parameter physical variability. Therefore, the choice of the likelihood function is crucial. In the present study the following function was used [32]:

$$L(Q/\Theta) = exp\left[\frac{-\sum_{i=1}^{N}(Q_i - \hat{Q}_i)^2}{\kappa \cdot \sigma^2}\right] \tag{2}$$

with $Q_i$ and $\hat{Q}_i$ standing for modeled and observed discharge in the time step $i$, $\sigma^2$ the variance of the model residua, $\kappa$ the factor used to control the variance of the *a posteriori* distribution. The value of the $\kappa$ factor was estimated in order to maximize the likelihood of the observations in the output discharge distribution, ensuring that 95% of observation points were enclosed by 95% confidence intervals. For the $\hat{Q}_i$ flow rates measured with the ultrasonic flow meter, MES1 (at S1, Figure 1) were used.

### 2.5. Rainfall Depth

In engineering practice, IDF curves are widely used in the dimensioning of sewer systems and design of objects located on it (overflows, reservoirs, etc.) [4,8,14,22]. These curves describe the relation between the duration of rainfall ($t_r$) and the frequency of its occurrence ($C$), which can be generally written with the relation:

$$J = f(t_r, C, \xi) \tag{3}$$

in which $\xi$—regional parameter differentiating the variability of the obtained curves, $J$—mean rain intensity in $dm^3 \cdot (ha \cdot s)^{-1}$, expressed as $J = 166.7 \cdot P_{max}$.

Due to a number of conditions, such as the genesis of rainfall, climatic conditions, land use, the length of rainfall measurements, etc., the relationships described by Equation (1) may take different forms [20,33,34]. Taking into account the above remarks and the climate conditions in Poland, which was reflected in a series of works [20,35] aimed at developing empirical rainfall models, the formula

of Bogdanowicz and Stachy [32] was used in the analyses to determine the maximum rainfall depth ($P_{max}$) for the assumed precipitation frequency:

$$P_{max} = 1.42 \cdot t_r^{0.33} + \xi(R, t_r) \cdot (-lnp)^{0.584} \tag{4}$$

where $t_r$—duration of rain (min), $p$—probability of rainfall exceeding ($p = 1/C$), $\xi(R, t_r)$—regional parameter determined depending on the location and duration of rainfall.

The above relationship was developed on the basis of 60 years of precipitation measurements in Poland and is now used in engineering practice [4,5]. For the region covered by the research, i.e., the Świętokrzyskie Voivodship and the city of Kielce, the values of parameter $\xi$ are determined on the basis of equations:

$$\xi(R.t_r < 120 \ min) = 4.693 \cdot \ln(t_r + 1) - 1.249 \tag{5}$$

$$\xi(R.t_r < 1080 \ min) = 2.223 \cdot \ln(t_r + 1) + 10.639 \tag{6}$$

In further considerations at the stage of reservoir dimensioning for $C$ = 2, 5, 10, the rainfall duration was considered in the range of $t_r$ = 15–240 min [9]. The results for $C$ = 1 were omitted in the analyses, as they lead to an underestimation of reservoir sizes, which was reported in various studies [4,5,9].

## 2.6. Surface Runoff Modelling

One of the factors determining the capacity of the reservoir and the type of drains designed is the outflow from the catchment. This outflow is the result of the runoff on the surface of the catchment area, and then of the flow of stormwater through the sewer system. In this paper, the SWMM model was used to simulate the drain from the catchment. This model is used commonly to simulate the quantity and quality of stormwater and the phenomenon of rainfall–runoff is modelled in it, taking into account non-linear reservoirs, where infiltration and surface runoff are simulated [36].

Due to the complex process of wastewater accumulation in the overflow reservoir, requiring large calculation outlays in the SWMM model, a simplified model for the reservoir was used in the conducted analyses [5,6,9]. In this model, the overflow reservoir capacity can be determined on the basis of the differential equation of the wastewater volume balance of the form:

$$\frac{dV(t)}{dt} = S \cdot \frac{dH(t)}{dt} = I(t) - Q(t)_{out} \tag{7}$$

where $S$—surface area of the reservoir in the projection, $H(t)$—reservoir depth, $I(t)$—inflow to the reservoir determined on the basis of SWMM simulations, $Q(t)_{out}$—outflow from the reservoir, with negligible error that can be assumed to be constant [36].

## 3. Results

On the basis of SWMM model calculation results and literature data, the ranges of variability of individual model parameters were determined (Table 1). Due to strong interactions between the calibrated parameters and their significant influence on the results of surface runoff calculations, the following modification coefficients were introduced at the calculation stage: coefficient ($\alpha$) for the flow path of width ($W_j$), coefficients ($d_{imp}$; $d_{perv}$) for the retention depth of impervious and pervious areas, coefficients ($n_{imp}$; $n_{perv}$) of roughness of impervious and pervious areas, coefficients ($n_n$) of roughness of sewer, coefficient ($\gamma$) for the share of impervious areas ($Imp_j$) in the whole catchment area, coefficient ($\beta$) for the average slope of a partial catchment ($I_j$). In order to reduce the multidimensionality of the solution and limit the outlays of calculations, it was assumed that the flow path width is determined as $W = \alpha \cdot A^{0.5}$. The values of coefficients ($\beta, \gamma$) are used to correct the value of the sealed area ($Imp_j$) and the slope of the catchment ($I = \beta \cdot I_j$), where $j$—the number of partial catchments.

**Table 1.** Ranges of variability of model parameters for the *a priori* uniform distribution [16].

| Parameters | Unit | Range |
|---|---|---|
| Coefficient for flow path width ($\alpha$) | - | 2.7–4.7 |
| Retention of impervious areas ($d_{imp}$) | mm | 0.8–4.8 |
| Retention of pervious areas ($d_{perv}$) | mm | 0.8–6.8 |
| Roughness coefficient for impervious areas ($n_{imp}$) | $m^{-1/3} \cdot s$ | 0.01–0.022 |
| Roughness coefficient for pervious areas ($n_{perv}$) | $m^{-1/3} \cdot s$ | 0.16–0.20 |
| Roughness coefficient of sewer channels ($n_n$) | $m^{-1/3} \cdot s$ | 0.01–0.048 |
| Correction coefficient for percentage of impervious areas ($\gamma$) | - | 0.7–1.275 |
| Correction coefficient for sub-catchments slope ($\beta$) | - | 0.8–1.375 |

The Monte Carlo sample was conditioned using a likelihood function (Equation (7)) on two sets of observations: for 15 September 2010 (time and total rainfall depth equal to $t_r$ = 107 min and $P_{tot}$ = 9.2 mm, volume of runoff $V_{tot}$ = 2221 $m^3$), and 7 August 2011 ($t_r$ = 60 min; $P_{tot}$ = 8.6 mm, volume of the runoff $V_{tot}$ = 1733 $m^3$), events [37]. The scaling factor $\kappa$ was adjusted in order to ensure that 95% of observations in calibration sets are enclosed in 95% confidence intervals. In Figure 3, resulting parameter distributions in respect of the likelihood measure (Equation (8)) are showed as box-plots. The total variability of the likelihood function is presented with dashed lines extents. A red center line denotes the median value and 25th and 75th quantiles are indicated by box edges. Box-plots are used instead of dot-plots, found in Kiczko et al. [27], as they provide a more precise characterization of the *a posteriori* distribution. For $\alpha$, $n_{perv}$, $n_{imp}$, $d_{perv}$, and $\gamma$, the model response is almost uniform—similar high and low likelihood values were found in the whole parameter span (Figure 3a–c,e,g). It is different in the case of remaining parameters, $d_{imp}$, $\beta$, and $n_n$ (Figure 3d,f,h). This suggests that the model is more sensitive to the second parameter set, as it is possible to spot the dependency of the likelihood function on parameter values. For example, it is clear that the highest model likelihood was obtained for $\beta$ values close to 0.9 (Figure 3f). Figure 4 presents verification of the probabilistic solution for an independent data set. On the basis of the simulations performed (Figure 4), it can be stated that the measured and modelled outflow hydrographs for the calibration set are characterized by a high adjustment for high intensity rainfall (time and total rainfall depth equal to $t_r$ = 270 min and $P_{tot}$ = 16.5 mm, which results in rainfall intensity $q$ = 10.19 $dm^3 \cdot (ha \cdot s)^{-1}$ at the volume of $V_{tot}$ = 3415 $m^3$ hydrograph).

Using the simulation results obtained with the Monte Carlo method and *a posteriori* distributions of the model parameters (Table 1), hydrograph calculations of outflow from the catchment were performed for precipitation $t_r$ = 10–240 min and $C$ = 1, 2, 5, 10, determined on the basis of IDF curves described with Equations (4)–(6). Next, on the basis of Equation (7) for the assumed design flows ($Q_{out,max}$ = 0.1–1.0 $m^3 \cdot s^{-1}$), capacities of retention reservoirs were determined, thus obtaining a probabilistic solution and determining for individual values of $V(t_r, C, q)$ the mean value and 95% confidence interval, respectively.

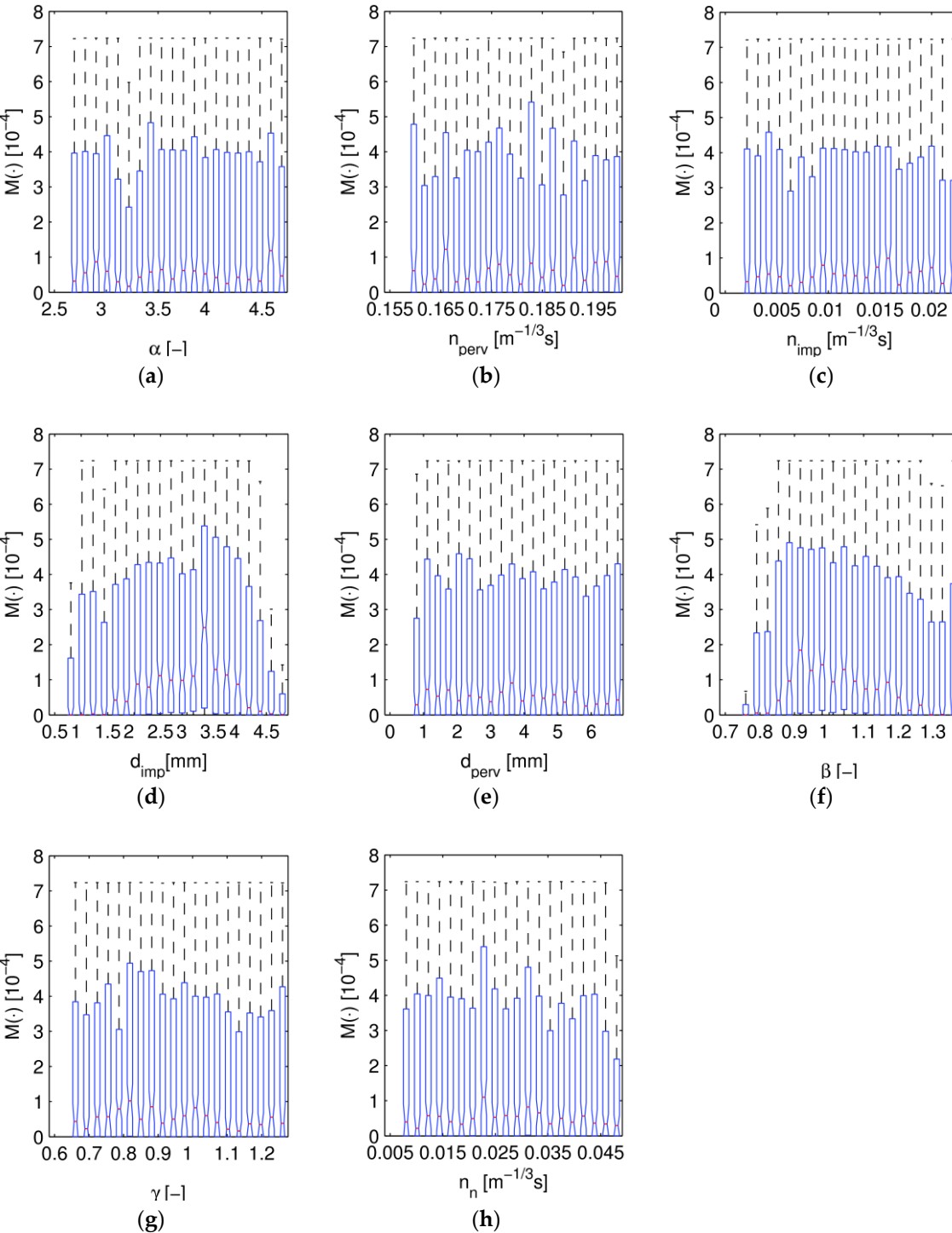

**Figure 3.** Box-plots for likelihood function values in the respect of parameter values (parameters symbols explained in Table 1).

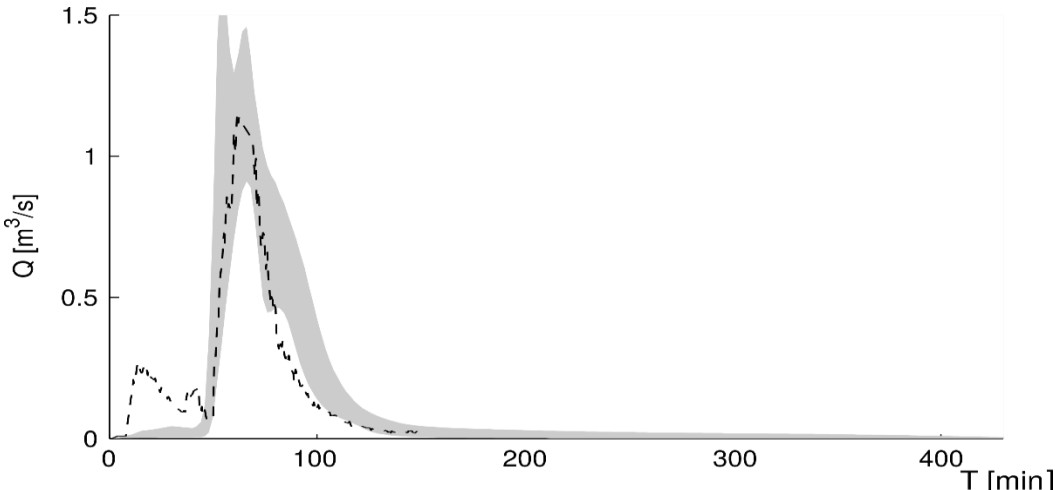

**Figure 4.** Exemplary results of calculations and measurements for the confidence interval of 95% using the Generalized Likelihood Uncertainty Estimation (GLUE) method for the hydrograph of outflow from the catchment (30 July 2010).

The results were compared with the reservoir capacities obtained neglecting the uncertainty, for a single parameter set, ensuring satisfactory simulation results. This approach is commonly used in engineering practice and is called the deterministic solution. The results of calculations obtained in this way allowed to assess the influence of uncertainty of the calibrated parameters of the SWMM model on the determined reservoir capacities.

Using the above remarks and based on simulations performed with the SWMM model, the influence of the uncertainty of the hydrodynamic model of the catchment and unit outflow on the design rainfall duration used to dimension the reservoir and the unit reservoir capacity was analyzed. On the basis of performed calculations, the dependence $V_q = f(t_r)$ for selected values of $q$ was first determined; an example of the dependence $V_q = f(t_r)$ for $C = 5$ and $q = 17$ dm$^3\cdot$(ha·s)$^{-1}$ and $q = 6$ dm$^3\cdot$(ha·s)$^{-1}$ is shown in Figure 5.

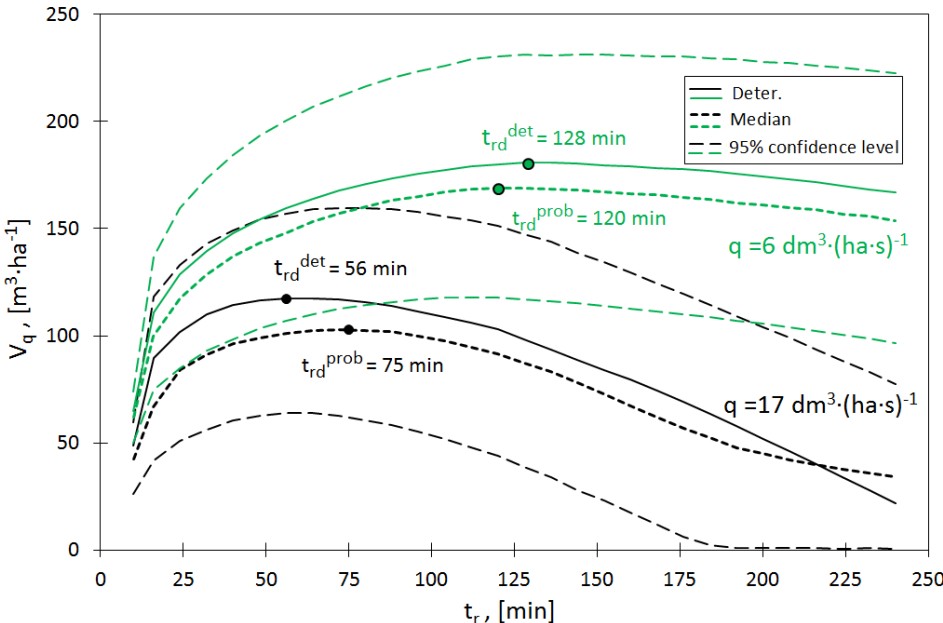

**Figure 5.** Influence of the unit outflow from the reservoir ($q$) and the time duration rainfall ($t_r$) on the unit accumulation capacity ($V_q$), taking into account the model uncertainty.

From the results obtained it can be concluded that the uncertainty of the hydrodynamic model of the catchment influences the unit accumulation capacity of the reservoir, which is confirmed by the range of variability of the confidence interval of 95% for individual values of $V_q = f(t_r)$ and by the dependence of $V_q = f(q, C)$ shown in Figure 8. Moreover, it was found that the reservoir capacity determined taking into account the model uncertainty (median) is smaller than the value obtained in the deterministic solution, which is also indicated by the variability of $V_q$ values illustrated in Figure 5. Due to the fact that curves in Figure 5 were prepared for selected values of $q$, it is impossible to generalize them for the remaining unit values of reservoir outflows. For this purpose, it was necessary to analyze the relations between unit reservoir capacities obtained for $t_{rd}^{prob}$ and $t_{rd}^{det}$ and the assumed outflows ($q$) and the assumed rainfall frequency ($C$), which is discussed in the further part of the paper.

On the basis of the curves shown in Figure 5 and determined for $q = 6$ dm³·(ha·s)$^{-1}$ and $q = 17$ dm³·(ha·s)$^{-1}$, it is possible to determine the influence of the SWMM model uncertainty on the duration of the design rainfall used to dimension the reservoir in the deterministic and probabilistic solutions. On the basis of the presented curves, it can be stated that the value of the design rainfall duration in the deterministic solution ($t_{rd}^{det}$) is shorter than the value obtained in the probabilistic one (median)—$t_{rd}^{prob}$. In the first solution for $q = 17$ dm³·(ha·s)$^{-1}$, the $t_{rd}$ value is 56 min and is 19 min shorter than in the second one, and the $\frac{t_{rd}^{prob}}{t_{rd}^{det}}$ ratio is 1.33, while for $q = 6$ dm³·(ha·s)$^{-1}$ the shift between rain duration in the considered solutions is shortened to 8 min and the $\frac{t_{rd}^{prob}}{t_{rd}^{det}}$ of 1.09 decreases accordingly. The obtained result indicates that the smaller the unit outflow from the reservoir ($q$), the value of the shift between the duration of rainfall obtained in deterministic and probabilistic solutions decreases, which translates into the size of the required storage capacity of the reservoir. Bearing in mind the need to carry out a detailed evaluation of the model uncertainty influence on individual aspects of reservoir capacity selection and its operation, on the basis of the obtained results the $\frac{t_{rd}^{prob}}{t_{rd}^{det}} = f(q, C)$ and $\frac{V_q^{prob}}{V_q^{det}} = f\left(\frac{t_{rd}^{prob}}{t_{rd}^{det}}, C\right)$ curves were prepared and shown in Figures 6 and 7.

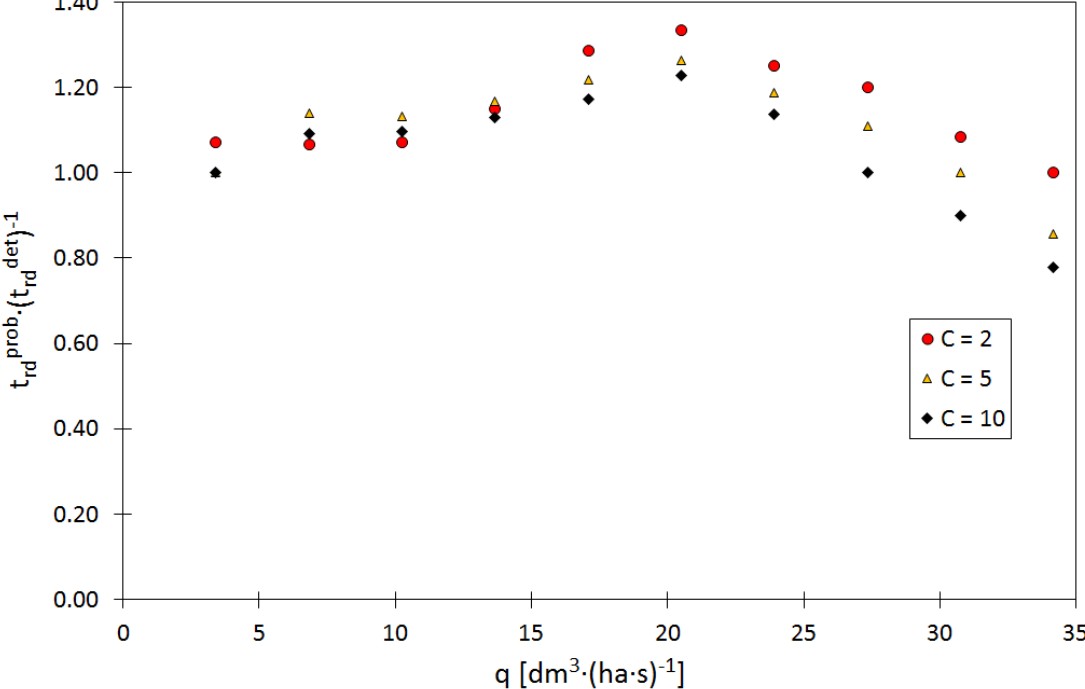

**Figure 6.** Influence of unit outflow from the reservoir ($q$) and rainfall frequency ($C$) on the difference between the design rainfall duration obtained for deterministic and probabilistic solutions.

On the basis of the data in Figure 6, it can be stated that the difference in the shift between the rain duration determined in the probabilistic and deterministic solutions for designing the retention reservoir is influenced by the unit outflow ($q$) and the rainfall frequency ($C$). The results of calculations shown in Figure 6 showed that the highest relative difference between the $t_{rd}$ time obtained in the deterministic and probabilistic solution was obtained for $q = 21$ dm$^3$·(ha·s)$^{-1}$ for $C = 2$–10, and it is the highest for $C = 2$ and exceeds 35%. However, for the value of $C = 10$, it is smaller than for $C = 5$ and $C = 2$, and is 1.15. A further increase in the unit value of the outflow from the reservoir ($q$) leads to a decrease in the time lag between the duration of rainfall obtained in the deterministic and probabilistic solutions. The issue discussed above is important from the point of view of the reservoir design, because identification of the most unfavorable distribution of $t_{rd}^{prob}$ values in relation to the deterministic solution defines the area of operation of the reservoir, in which it is most exposed to potential underestimation.

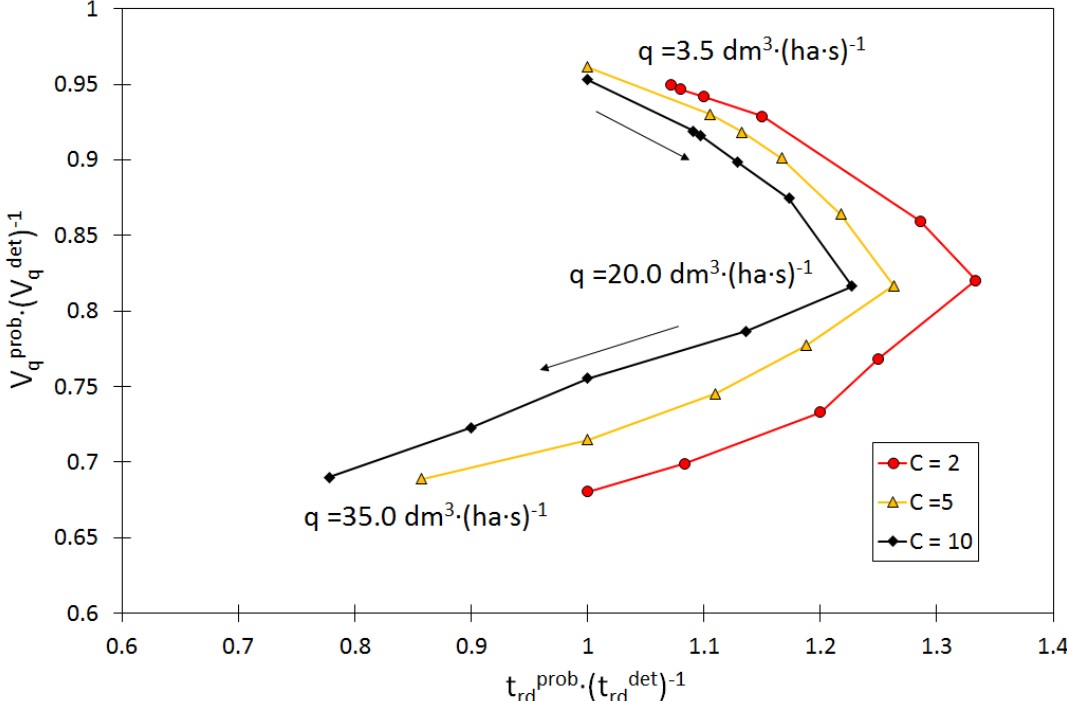

**Figure 7.** Influence of relative shift between probabilistic and deterministic solution and unit reservoir capacities ($V_q$).

Taking into account the above considerations, the variability of $\frac{V_q^{prob}}{V_q^{det}} = f\left(\frac{t_{rd}^{prob}}{t_{rd}^{det}}, C\right)$, allowing for determination of the effect of the relative shift between the $t_{rd}$ value determined for deterministic and probabilistic solutions on the unit storage capacity, is presented in Figure 7. Analyzing the shape of the curves, it can be stated that the increase in the relative shift between the design rainfall duration obtained in the deterministic and probabilistic solution (median) relative to the maximum value of $\frac{t_{rd}^{prob}}{t_{rd}^{det}}$ corresponding to $q = 15$ dm$^3$·(ha·s)$^{-1}$ ($C = 2$–10) leads to a decrease in the relative difference in the capacity of reservoirs obtained in the SWMM model and taking into account the uncertainty, and reaches the value equal to $\frac{V_q^{prob}}{V_q^{det}} = 0.8$. As a result of a decrease in the $\frac{t_{rd}^{prob}}{t_{rd}^{det}}$ value, which is related to the increase in the outflow from the reservoir (Figure 5), the $\frac{V_q^{prob}}{V_q^{det}}$ quotient decreases further, the value of which indicates an underestimation of the storage capacity in the probabilistic solution (the median is a search solution) in relation to the value obtained only by the calibrated SWMM model.

In the next stage of analyses, on the basis of calculations carried out with the SWMM program, curves were determined (Figure 8) showing the influence of the unit outflow from the reservoir ($q$) on the unit maximum accumulation capacity ($V_q$) at preset values $C$ for the design rainfall duration ($t_{rd}$). The calculations done showed that the size of the unit storage capacity of the reservoir ($V_q$) determined in the deterministic solution is larger than the median obtained in the probabilistic solution, and its value decreases with the increase of the unit outflow from the reservoir, which is also confirmed by the variability of $V_q = f(t_r, C)$ in Figure 8. This result indicates that the unit storage capacity of the reservoir ($V_q$) obtained in the deterministic solution is overestimated in relation to the capacity obtained in the probabilistic solution (median), which means that the reservoir may be oversized. From the point of view of the reliability of the reservoir operation, this is a beneficial solution, however, taking into account the 95% confidence interval, it is difficult to draw far-reaching generalizations, as it may turn out that the capacity of the reservoir designed on the basis of a deterministic solution may still be insufficient.

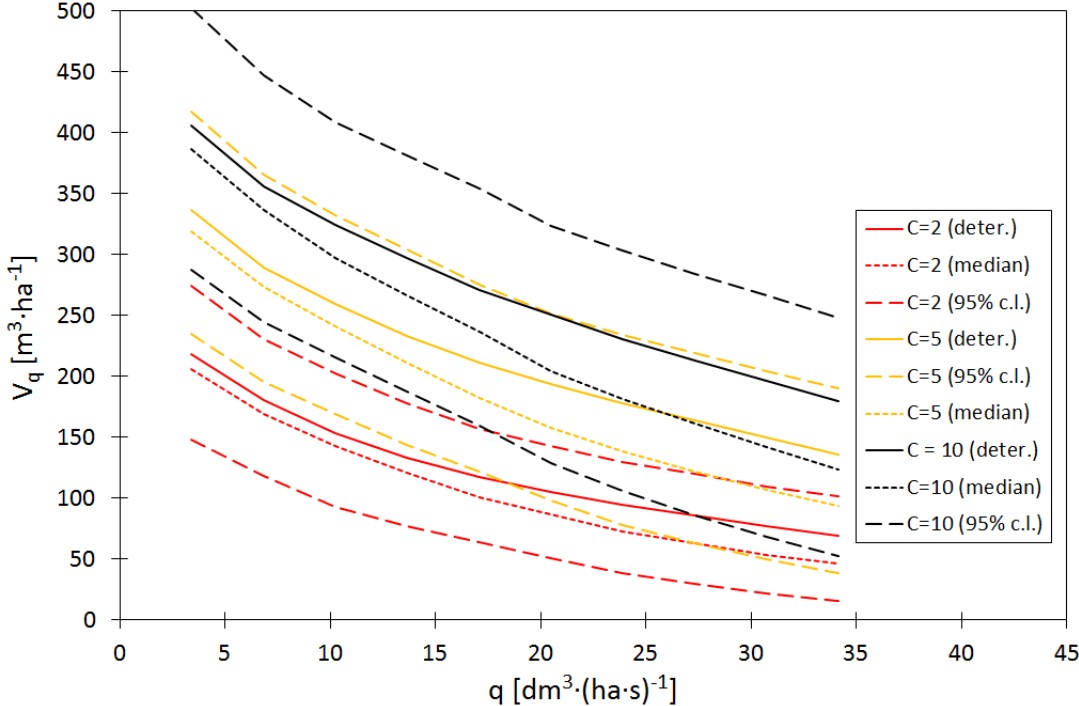

**Figure 8.** Influence of the unit outflow from the reservoir (q) and the rainfall frequency (C) on the unit accumulation capacity ($V_q$), taking into account the model uncertainty (where: c.l.—confidence level).

Therefore, further analyses are necessary in this respect, all the more so as retention reservoirs are important objects on the stormwater networks and the appropriate selection of their dimensions is of key importance for the high reliability of the drainage systems operation.

The analysis of the obtained curves $V_q = f(q, C)$ indicates an increase in the relative unit difference of reservoir capacities obtained in deterministic and probabilistic solutions, depending on the value of $q$. For example, for $C = 2$ an increase in the value from $q = 6.8$ dm$^3\cdot$(ha$\cdot$s)$^{-1}$ to $q = 13.7$ dm$^3\cdot$(ha$\cdot$s)$^{-1}$ leads to an increase in the relative difference in reservoir capacity from 6% to 19%. Referring the obtained result to the above analyses, it can be stated that with the increase in the amount of outflow from the reservoir, the degree of its oversizing (taking the probabilistic solution as the starting point) increases, which from the point of view of economy and operation of the stormwater system below, the reservoir is unfavorable.

Based on Figure 8, it can be concluded that an increase in the unit outflow from the reservoir ($q$) leads to a reduction in the required unit accumulation capacity ($V_q$). This means that the uncertainty of

the model is important for the selection of discount devices. Comparing the values of unit outflows from the reservoir for the assumed unit accumulation capacity, it can be stated that for the deterministic solution the outflows, *q* are greater than taking into account the uncertainty. This is important when designing the reservoir, as omitting the uncertainty leads to an overstatement of the outflow with the drain, which may lead to unfavorable phenomena (system overload) in the stormwater system located below the outlet from the reservoir.

## 4. Summary and Conclusions

The paper presents a methodology of reservoir dimensioning taking into account the uncertainty of identification of parameters calibrated in the hydrodynamic model. The calculations made in the paper showed that the uncertainty of the model has a significant impact on the design of the retention reservoir. The simulations showed that due to the increase of the unit outflow from the reservoir to *q* = 20 dm$^3 \cdot$(ha$\cdot$s)$^{-1}$, the design rainfall duration taken into account when dimensioning the reservoir (probabilistic solution) is longer than in the deterministic solution. Simultaneously, with the increase of *q*, the difference in the values of the design rainfall duration determined in the probabilistic and deterministic solutions increases. On the other hand, after exceeding a certain limit, a further increase in *q* leads to a decrease in the difference between precipitation duration obtained in the probabilistic and deterministic solutions.

On the basis of the analyses carried out, it was found that in the case under consideration, the unit storage capacity of the reservoir (median) obtained taking into account the uncertainty is smaller than that determined in the probabilistic solution. At the same time, it was shown that with the increase in unit outflow (*q*), the absolute difference in reservoir capacity (median) obtained in deterministic and probabilistic solutions increases. Therefore, the reservoir capacity (median value) determined on the basis of the hydrodynamic model simulation, where the uncertainty of the model was omitted, is underestimated compared to the deterministic solution. From the point of view of reliability of reservoir operation, this is a positive result, as the reservoir will not be overfilled. However, due to the range of variability of the established 95% confidence interval and the fact that these analyses were performed for a single catchment, it is difficult to draw far-reaching guidelines and generalizations. Therefore, further analyses for urban catchments with diversified physico-geographical characteristics are advisable.

**Author Contributions:** Conceptualization, B.S.; Methodology, B.S. and A.K.; Software, A.K. and B.S.; Writing—Original Draft, B.S., A.K., L.D.

**Funding:** The work was founded by the Polish Ministry of Science and Higher Education, the RID project, according to the agreement: 025/RID/2018/19 of 28/12/2018 with total budget of 12,000,000 PLN.

**Conflicts of Interest:** The authors declare no conflict of interest.

## Abbreviations

| | |
|---|---|
| *A* | subcatchment area in the runoff model, ha; |
| *C* | rainfall frequency; |
| $d_{imp}$ | retention depth of impervious areas in the runoff model, mm; |
| $d_{perv}$ | retention depth of pervious areas in the runoff model, mm; |
| $F_{imp}$ | impermeable of the catchment, ha; |
| $H(t)$ | reservoir depth, m; |
| $H_{max}$ | maximum reservoir depth, m; |
| $I(t)$ | inflow to reservoir, m$^3 \cdot$s$^{-1}$; |
| *I* | subcatchments slope in the runoff model; |
| *Imp* | percentage impervious areas in the runoff model; |
| $L(Q/\Theta)$ | likelihood function; |
| $n_{imp}$ | roughness coefficient for impervious areas in the runoff model, m$^{-1/3} \cdot$s; |

| | |
|---|---|
| $n_n$ | roughness coefficient for sewer channels in the runoff model, $\mathrm{m}^{-1/3}\cdot\mathrm{s}$; |
| $n_{perv}$ | roughness coefficient for pervious areas in the runoff model, $\mathrm{m}^{-1/3}\cdot\mathrm{s}$; |
| $P(\Theta)$ | denotes *a priori* parameter distribution; |
| $P(Q/\Theta)$ | the *a posteriori* distribution; |
| $P_{max}$ | maximum rainfall depth, mm; |
| $P_{tot}$ | rainfall depth in an episode, mm; |
| $p$ | probability of rainfall exceeding; |
| $Q_{out}$ | outflow from stormwater reservoir, $\mathrm{m}^3\cdot\mathrm{s}^{-1}$; |
| $Q_{out,max}$ | maximum outflow from stormwater reservoir, $\mathrm{m}^3\cdot\mathrm{s}^{-1}$ |
| $q$ | unit outflow from reservoir; $\mathrm{dm}^3\cdot(\mathrm{ha}\cdot\mathrm{s})^{-1}$ |
| $S$ | surface area of the reservoir in the projection, $\mathrm{m}^2$; |
| $t_r$ | rainfall duration, min; |
| $t_{rd}^{det}$ | design rainfall event—deterministic solution, min; |
| $t_{rd}^{prob}$ | design rainfall event—probabilistic solution, min; |
| $V_q$ | unit capacity index, $\mathrm{m}^3\cdot\mathrm{ha}^{-1}$; |
| $V_q^{det}$ | unit capacity index—deterministic solution, $\mathrm{m}^3\cdot\mathrm{ha}^{-1}$; |
| $V_q^{prob}$ | unit capacity index—probabilistic solution, $\mathrm{m}^3\cdot\mathrm{ha}^{-1}$; |
| $V_{tot}$ | volume of runoff, $\mathrm{m}^3$; |
| $W$ | flow path width in the runoff model, m; |
| $\alpha$ | coefficient for flow path width in the runoff model; |
| $\beta$ | correction coefficient for the percentage of impervious areas; |
| $\gamma$ | correction coefficient for subcatchments slope in the runoff model; |
| $\kappa$ | the factor used to control the variance of the *a posteriori* distribution. |

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
