# Peer review of "Stormwater Reservoir Sizing in Respect of Uncertainty"

_water, doi:10.3390/w11020321_

Round 1
Reviewer 1 Report
General Comments:
The authors present an interesting work which evaluates the impacts of hydrologic uncertainty when sizing a control reservoir for runoff. I think the paper is interesting way of looking at the problem and I agree with the overall methodology.
My primary concerns are related to the paper itself. First, the introduction does very little to establish the current state of the literature. There is much work on stochastic rainfall generation, hydrologic uncertainty, and methods for sizing control volumes which precede this paper. The introduction should be rewritten so that it is clear exactly how this paper is adding to the available literature. When I first read the introduction I thought this paper wouldn’t make a valuable contribution, but after finishing it I had changed my mind. I think the authors could help reach a broader audience by rewriting the introduction.
Second, the methodology is poorly organized. The study site and problem are introduced at the end. For example, it was very unclear why the GLUE method was being described and only later did I find out why.
Finally, many important details on the model calibration are missing from the methodology. What was measured and where? With what instrument? Can you provide some more standard calibration metrics (e.g. RMSE) to let us know how well the model performs?
Specific Comments:
Line 34: This first sentence is poorly structured. Please define “receiver.” Do you mean receiving water body?
First paragraph: These are global problems you describe, yet your focus is only on European requirements. I would broaden the geographic scope of this issue.
Line 44: Other approaches for sizing reservoirs have been considered. For example, see the recent work by Wang & Guo (2018).
Wang, J., & Guo, Y. (2018). An Analytical Stochastic Approach for Evaluating the Performance of Combined Sewer Overflow Tanks. Water Resources Research, 54(5), 3357-3375.
Line 71: The definition of rainfall characteristics in stochastic rainfall generation is important to the central aim of this paper. The authors should perform a more thorough review of this topic. For example, consider these papers as a starting point:
Knighton, J. O., & Walter, M. T. (2016). Critical rainfall statistics for predicting watershed flood responses: rethinking the design storm concept. Hydrological processes, 30(21), 3788-3803.
Vandenberghe, S., Verhoest, N., Buyse, E., & De Baets, B. (2010). A stochastic design rainfall generator based on copulas and mass curves. Hydrology and Earth System Sciences, 14(12), 2429-2442.
Vernieuwe, H., Vandenberghe, S., De Baets, B., & Verhoest, N. (2015). A continuous rainfall model based on vine copulas. Hydrology and Earth System Sciences, 19(6), 2685-2699.
Line 133: How are you handling the uncertainty in the SWMM runoff parameters? This has been identified as a critical issue in previous research on SWMM, particularly in designing detention basins:
Knighton, J., White, E., Lennon, E., & Rajan, R. (2014). Development of probability distributions for urban hydrologic model parameters and a Monte Carlo analysis of model sensitivity. Hydrological processes, 28(19), 5131-5139.
Knighton, J., Lennon, E., Bastidas, L., & White, E. (2016). Stormwater detention system parameter sensitivity and uncertainty analysis using SWMM. Journal of Hydrologic Engineering, 21(8), 05016014.
Sun, N., Hong, B., & Hall, M. (2014). Assessment of the SWMM model uncertainties within the generalized likelihood uncertainty estimation (GLUE) framework for a high‐resolution urban sewershed. Hydrological processes, 28(6), 3018-3034.
Zheng, Y., & Keller, A. A. (2006). Understanding parameter sensitivity and its management implications in watershed‐scale water quality modeling. Water Resources Research, 42(5).
Line 150: Not sure what you mean by “averaging.”
Line 153: I’m a little confused by the mention of a treatment plant here which was not previously introduced.
Line 166: Uncertainty of what? The uncertainty in defining rainfall characteristics, or hydrologic model parameters? Both?
Line 184: I would reorganize the paper to include this detail after the introduction. You go through detailed methodology to this point and it’s not entirely clear what the goal is.
Line 210: The calibration is not mentioned previously. This must be described.
Line 222: Your calibration data needs to be described in detail earlier in the paper. I’m not clear on what you are actually calibrating to at this point.
Author Response
Reviewer 1
(Comment 1)
My primary concerns are related to the paper itself. First, the introduction does very little to establish the current state of the literature. There is much work on stochastic rainfall generation, hydrologic uncertainty, and methods for sizing control volumes which precede this paper. The introduction should be rewritten so that it is clear exactly how this paper is adding to the available literature. When I first read the introduction I thought this paper wouldn’t make a valuable contribution, but after finishing it I had changed my mind. I think the authors could help reach a broader audience by rewriting the introduction.
First paragraph: These are global problems you describe, yet your focus is only on European requirements. I would broaden the geographic scope of this issue.
Line 44: Other approaches for sizing reservoirs have been considered. For example, see the recent work by Wang & Guo (2018).
Wang, J., & Guo, Y. (2018). An Analytical Stochastic Approach for Evaluating the Performance of Combined Sewer Overflow Tanks. Water Resources Research, 54(5), 3357-3375.
Line 71: The definition of rainfall characteristics in stochastic rainfall generation is important to the central aim of this paper. The authors should perform a more thorough review of this topic. For example, consider these papers as a starting point:
Knighton, J. O., & Walter, M. T. (2016). Critical rainfall statistics for predicting watershed flood responses: rethinking the design storm concept. Hydrological processes, 30(21), 3788-3803.
Vandenberghe, S., Verhoest, N., Buyse, E., & De Baets, B. (2010). A stochastic design rainfall generator based on copulas and mass curves. Hydrology and Earth System Sciences, 14(12), 2429-2442.
Vernieuwe, H., Vandenberghe, S., De Baets, B., & Verhoest, N. (2015). A continuous rainfall model based on vine copulas. Hydrology and Earth System Sciences, 19(6), 2685-2699.
Line 133: How are you handling the uncertainty in the SWMM runoff parameters? This has been identified as a critical issue in previous research on SWMM, particularly in designing detention basins:
Knighton, J., White, E., Lennon, E., & Rajan, R. (2014). Development of probability distributions for urban hydrologic model parameters and a Monte Carlo analysis of model sensitivity. Hydrological processes, 28(19), 5131-5139.
Knighton, J., Lennon, E., Bastidas, L., & White, E. (2016). Stormwater detention system parameter sensitivity and uncertainty analysis using SWMM. Journal of Hydrologic Engineering, 21(8), 05016014.
Sun, N., Hong, B., & Hall, M. (2014). Assessment of the SWMM model uncertainties within the generalized likelihood uncertainty estimation (GLUE) framework for a high‐resolution urban sewershed. Hydrological processes, 28(6), 3018-3034.
Zheng, Y., & Keller, A. A. (2006). Understanding parameter sensitivity and its management implications in watershed‐scale water quality modeling. Water Resources Research, 42(5).
Line 184: I would reorganize the paper to include this detail after the introduction. You go through detailed methodology to this point and it’s not entirely clear what the goal is.
Answer 1
We are very grateful for Reviewers remarks. We revised the manuscript according to reviewer’s recommendations. Now, the introduction provides the current state-of-art for reservoir sizing in stormwater systems. The problem of the precipitation modeling was also described, including references to studies based on multidimensional probability distributions, approximated with kernel functions, as suggested by the Reviewer. We also referred to other methods (like Iman-Conover and canonical microcascades) used for simulations of rainfall time series. The IDF approach (Intensity – Duration – Frequency) for rainfall simulations was also explained.
The revised manuscript includes additional references to studies on uncertainty analysis of runoff models.
Comment 2
Second, the methodology is poorly organized. The study site and problem are introduced at the end. For example, it was very unclear why the GLUE method was being described and only later did I find out why.
Answer 2
We reorganized the structure of the manuscript. In the present version, we started from the description of the study area, followed by methodology: (1) an estimation method for the duration of the design rainfall and the calculation of the capacity (2) assessment of the model uncertainty using GLUE approach; (3) the approach for determining the design rainfall height in respect of its duration and frequency; (4) calculation of the basin outflow hydrograph.
Comment 3
Finally, many important details on the model calibration are missing from the methodology. What was measured and where? With what instrument? Can you provide some more standard calibration metrics (e.g. RMSE) to let us know how well the model performs?
Answer 3
The measurement devices and their localization are described in the study area section. The stormwater flow rate and water level were measured with 1 minute time step. The manuscript addresses also fit measures (Eq. 2) of the simulated outflow hydrograph. Please note however, that measures like RMSE are not straightforward with a probabilistic model (or with the identification problem formulated in a probabilistic manner, like the GLUE approach). Here, the calibration is considered as successful, if uncertainty bands enclose observations.
Comment 4
Specific Comments:
Line 34: This first sentence is poorly structured. Please define “receiver.” Do you mean receiving water body?
Line 150: Not sure what you mean by “averaging.”
Line 153: I’m a little confused by the mention of a treatment plant here which was not previously introduced.
Line 166: Uncertainty of what? The uncertainty in defining rainfall characteristics, or hydrologic model parameters? Both?
Line 210: The calibration is not mentioned previously. This must be described.
Line 222: Your calibration data needs to be described in detail earlier in the paper. I’m not clear on what you are actually calibrating to at this point.
Answer 4
We included remarks in the revised version of the manuscript.
Reviewer 2 Report
You can find my comments on the article.

Author Response
Comment 1
The 84 GLUE method was used for uncertainty analysis. Calculations of reservoir capacity were made on 85 the basis of IDF curves developed for Poland.” you should make definition first?
Answer 1
The abbreviation GLUE (Generalized Likelihood Uncertainty Estimation) is now explained in the introduction.
Comment 2
Calculation diagram of the method of dimensioning the retention capacity”. Many symbols, not clear.
Answer 2
Symbols used in the figure demonstrating the approach of reservoir sizing, were explained.
Comment 3
In further considerations at the stage of reservoir dimensioning for ?=2,5,10 the rainfall duration was 130 considered in the range of ??=15÷240 min.” based on what?
Review 3
We provided explanations, including references, why simulations were limited to precipitation duration times n the range of 15-240 min.
Comment 4
Study area add more about rainfall, temp, evt....etc.
Answer 4.
We provided more details on the case study, including population density, road network density, variability of precipitation and temperature, number of snow days.
Answer 5
The main channel receives rainwater from 17 side channels with diameters of 300÷1000 mm.”
We revised the sentence: The main channel receives rainwater from 17 side channels with diameters of 300÷1000 mm.”, which now sounds:
“The main channel receives rainwaters from 17 side channels, whose diameters are varying in the range of 300÷1000 mm”
Round 2
Reviewer 1 Report
The authors have addressed my major concerns. I support this article for publication.